# IgY Antibodies from Birds: A Review on Affinity and Avidity

**DOI:** 10.3390/ani13193130

**Published:** 2023-10-07

**Authors:** Bianca Lisley Barboza Pacheco, Camila Parada Nogueira, Emerson José Venancio

**Affiliations:** 1Scientific Initiation Programme, Biomedicine Course, State University of Londrina, Londrina 86038-350, Brazil; biancalisley90@gmail.com; 2Scientific Initiation Programme, Animal Science Course, State University of Londrina, Londrina 86038-350, Brazil; camila.paradan97@uel.br; 3Department of Pathological Sciences, State University of Londrina, Londrina 86038-350, Brazil

**Keywords:** chicken IgG, immunoglobulin Y, affinity maturation, immunochemistry

## Abstract

**Simple Summary:**

IgY antibodies are used in research and in the development of solutions for immunotherapy and the immunodiagnosis of human and animal diseases. Affinity and avidity are forces that describe the interaction between an antigen and antibody and are important characteristics for the biological function of IgY antibodies. Therefore, these measures are fundamental variables for the development of immunodiagnostic methodologies and immunotherapy based on IgY antibodies. In this review, we address factors that influence the affinity and avidity of IgY antibodies and the methodologies used for the determination of these strengths. We observed a low number of studies on the factors influencing the maturation of IgY affinity and avidity and a wide variation in the methodologies used to determine these variables. The development of studies characterising the factors that influence the maturation of IgY antibody affinity and avidity, with standardised methodologies for the determination of these forces, is of utmost importance.

**Abstract:**

IgY antibodies are found in the blood and yolk of eggs. Several studies show the feasibility of utilising IgY for immunotherapy and immunodiagnosis. These antibodies have been studied because they fulfil the current needs for reducing, replacing, and improving the use of animals. Affinity and avidity represent the strength of the antigen–antibody interaction and directly influence antibody action. The aim of this review was to examine the factors that influence the affinity and avidity of IgY antibodies and the methodologies used to determine these variables. In birds, there are few studies on the maturation of antibody affinity and avidity, and these studies suggest that the use of an adjuvant-type of antigen, the animal lineage, the number of immunisations, and the time interfered with the affinity and avidity of IgY antibodies. Regarding the methodologies, most studies use chaotropic agents to determine the avidity index. Studies involving the solution phase and equilibrium titration reactions are also described. These results demonstrate the need for the standardisation of methodologies for the determination of affinity and avidity so that further studies can be performed to optimise the production of high avidity IgY antibodies.

## 1. Introduction

Avidity is a key measure of the strength of the interaction between antigen and antibodies and plays a key role in antibody function [1]. The higher the avidity, the longer the interaction time of the antigen with the antibody, and the more likely the antibody is to trigger the biological reactions necessary for the elimination of the antigen [2]. The increase in avidity throughout the development of the humoral immune response is a characteristic of this response and an area of intense research [3]. Understanding the molecular process involved in increased avidity is of fundamental importance, especially in vaccine development. In poultry, there are few studies on the avidity and affinity of IgY antibodies. These antibodies are equivalent to mammalian IgG antibodies and are the most abundant in serum, and their levels increase as the humoral immune response develops [4]. Currently, IgY antibodies have been used to develop applications for immunotherapy and the immunodiagnosis of diseases in humans and animals [5]. Despite their many advantages over mammalian antibodies, there are few IgY-based products available on the market. Understanding the mechanisms involved in affinity maturation can result in the establishment of immunisation protocols that lead to the production of high-avidity IgY antibodies in the shortest possible time. Consequently, this can increase the competitiveness of these antibodies compared to those produced in mammals. In the current work, we review the studies that have investigated factors that affect the affinity and avidity of IgY antibodies and the methodologies used to determine these variables.

## 2. General Characteristics of IgY Antibodies

IgY antibodies are one of the classes of immunoglobulins found in birds [5]. They were initially called chicken IgG or 7S antibodies due to their similarities to mammalian IgG antibodies [6]. These avian Y antibodies are related to the IgY antibodies found in reptiles and amphibian birds [7,8]. They are found in blood and tissues and are transferred from the circulatory stream to the developing yolk via specific receptors, where they are stored and have the function of protecting the embryo [9]. In blood and yolk, the concentration of IgY antibodies is variable and influenced by factors such as breed, age, and antigenic stimulation [10,11,12]. Values between 4 and 14 mg/mL have been described in blood, while values between 7 and 15 mg/mL have been observed in yolk [10,11,12]. Interestingly, there is a direct proportional correlation between IgY antibody levels in serum and yolk [13]. To date, no significant differences have been described between IgY antibodies found in serum or yolk, either in the structure or in their characteristics, such as an antigen-binding capacity or avidity [14,15]. These immunoglobulins have a similar role to mammalian IgG. They are produced in higher concentrations in the secondary immune response, acting as opsonins and being involved in the activation of the complement system via the classical pathway [16,17]. The molecular structure of IgY antibodies is similar to that of other immunoglobulins. The IgY molecule is composed of two larger amino acid chains, the so-called heavy chains (HCs), and two smaller chains, the so-called light chains (LCs), and has an estimated molecular mass of approximately 170 kDa [18]. The HCs are joined via disulfide bridges and each HC is also joined to a light chain via a disulfide bridge. The HCs are composed of five immunoglobulin domains named, in the direction from the amino terminal end to the carboxy terminal end, the HC variable domain, the 1st HC constant domain, the 2nd HC constant domain, the 3rd HC constant domain, and the 4th HC constant domain. The molecular mass of the HC is estimated to be approximately 65 kDa [18]. LCs are composed of two domains, called the LC variable domain (amino-terminal region) and the LC constant domain (carboxy-terminal region), and are approximately 18 kDa [18]. As in other immunoglobulins, the antigen-binding site is formed via the the juxtaposition of the LC variable domain and the HC variable domain, in which the positions of great amino acid diversity are found, called complementarity determining regions (CDR1, CDR2, and CDR3). These positions are very important for antibody avidity [16]. The IgY molecule has two identical combinatorial sites and is considered a bivalent antibody. A detailed description of the molecular structure and genes of IgY antibodies can be found in another review [16].

## 3. Applications

IgY antibodies are molecules of great interest for immunotherapy, immunodiagnosis, and basic research [19,20,21,22,23,24]. The production of IgY antibodies fulfils the current need for reducing, replacing, and improving the use of animals, since IgY antibodies can be produced in laying hens instead of using mammals, leading to less exposure to suffering and a significant reduction in the number of animals used [25]. This is possible because IgY antibodies can be obtained directly from the egg yolk of laying hens and other birds via relatively simple and low-cost purification methods [26,27]. This avoids the need for bleeding or slaughtering the animals used for antibody production [23]. In addition, one egg yolk can yield an additional 100 mg of IgY antibodies, and, considering that laying hens produce almost 30 eggs per month, a single hen can replace several rabbits in antibody production [23].

Immunotherapy studies show the possible use of IgY antibodies for the prevention and treatment of diseases in humans and animals [17,28,29,30,31,32]. In particular, IgY antibodies have been studied for immunotherapy of bacterial [12,33], viral [29,34], fungal [30,35], parasitic [31,36], respiratory [37], enteric [38,39,40], and chronic diseases, such as periodontitis, cystic fibrosis, and coeliac disease [41,42,43]. Within the context of immunotherapy, which is different from the antibodies produced in mammals, IgY antibodies can be utilised without the need for processing to remove the Fc portion. This is possible because IgY antibodies do not activate the complement system or interact with mammalian Fc receptors, which makes them safe for immunotherapy in mammals [17,23].

The development of IgY-based immunodiagnostic reactions is an area of intense research [17,31,44,45], and reviews on the application of IgY antibodies in the diagnosis of infectious and chronic diseases have been published [34,36,46]. IgY antibodies have been used in the development of ELISA, Western blotting, immunohistochemistry, immunochromatography, immunofluorescence, radioimmunoassay, and biosensors for the diagnosis of infectious and chronic diseases in humans and animals [46,47,48]. The use of IgY antibodies presents some advantages over mammalian antibodies, the most important of which are the non-interaction with the rheumatoid factor or mouse anti-IgG antibodies, with consequent interference in the test results [49,50]; the non-activation of the complement system and the generation of its fragments, which may result in the covering of epitopes important for diagnosis [49]; and a higher molecular stability than mammalian antibodies [51].

In basic research, IgY antibodies are widely used. In particular, due to the phylogenetic distance between birds and mammals, birds allow the production of specific antibodies against the antigens conserved in mammals [23,24,52,53]. In addition, the fact that they can be produced on a large scale enables the production of antibodies to meet the need for the characterisation of proteins identified using genomic studies [54]. Finally, IgY antibodies have been used to develop products for the optimisation of proteomics analyses [55].

In addition to these broad areas of application, studies have shown the application of IgY antibodies in the areas of food preservation, bioterrorism, and genetically modified organisms’ detection [23,56,57,58].

The main difficulties for the more intensive utilisation of IgY antibodies are probably their sensitivity to the acidic pH of the stomach, low efficacy against gram positives, higher production cost compared to antibiotic production, low half-life in mammals, low bioavailability, and concerns regarding the development of allergic reactions because IgY antibodies are egg-derived molecules [29,59,60].

## 4. Affinity and Avidity

Regardless of the different uses of IgY antibodies, as with other antibodies, their main function is to interact with the antigen. This interaction depends primarily on the combinatorial site of the antibody (ab)—the region formed via the union of the variable regions of the HCs and LCs—and the epitope present on the antigen (ab). The Fc portion of the antibody may also contribute to the ab–ag interaction [61]. This interaction and its duration are related to a set of non-covalent forces that are inversely proportional to distance, such as ionic forces, hydrogen bridges, hydrophobic forces, and van der Walls forces [62,63]. These forces are stronger when the distance between the elements is smaller. Therefore, the intensity of these forces is dependent on the complementarity between the antigen and the antibody. The greater the complementarity of the antigen–antibody interaction, the greater the binding force between them. The expression of the interaction force between one epitope and one combinatorial site is called affinity. A key feature of affinity is that it is variable during the development of a specific immune response, with an increase in antibody affinity observed throughout contact with the antigen or with repeated contact with the same antigen [64,65]. This process is known as affinity maturation [3,66]. Affinity can be expressed via the association constant at equilibrium (K) or via the dissociation constant (Kd or Kdiss) [1].

Several studies show that affinity increases 10- to 100-fold over the course of the specific immune response, and the mechanisms involved in affinity maturation are the subject of intense research [3,66,67]. In mammals, it is well established that the process of affinity maturation occurs in germinal centres. Cellular structures where B lymphocytes undergo the process of somatic hypermutation result in changes in antibody variable regions and the selection of antibody-producing B lymphocytes with higher affinity [3]. This process occurs during an intense migration of B lymphocytes between the dark zone and the light zone present in the germinal centres. The dark zone is an area within the germinal center where numerous B lymphocytes are actively dividing and undergoing somatic hypermutation. In contrast, the light zone contains fewer cells and is responsible for stimulating the survival of B lymphocytes using various processes, the expression of antibodies with greater avidity, and the death by apoptosis of unselected lymphocytes [68]. These processes involve follicular dendritic cells and follicular T lymphocytes [68]. Experimental evidence suggests that similar processes occur in the germinal centres of birds [69,70]. The germinal centre found in chickens is formed via an outer region with intense cell proliferation and where somatic hypermutation occurs [71,72], and an inner area where follicular dendritic cells are present [73]. In addition, the presence of CD3+ cells, the class change from IgM to IgY and the occurrence of apoptosis have been described in chicken GC [74,75]. An important observation is a slower affinity maturation in chickens than in rabbits [76]. The authors attribute this to the smaller number of variable regions in birds compared to mammals; however, this result is the opposite to that observed by another study [15]. In any case, there are few studies on the process of affinity maturation in these animals.

An important feature is that affinity does not fully describe the interaction between the antigen and antibody. Considering that an antigen can have more than one copy of the same epitope—i.e., have a valence greater than 1, and the antibody has at least two identical antigen-binding sites—and is therefore at least bivalent, the strength of the antigen–antibody interaction will depend on the valence of these molecules [2]. The role of the antigen and antibody valence in the strength of the antigen–antibody interaction is measured using avidity. Avidity is influenced by affinity, the valences of the antibody and the antigen, and the geometry of the interaction between the antigen and antibody [1,2]. Avidity can also be expressed in terms of the constants K and Kd [1]. It is important to emphasise that in the literature, the terms avidity and antibody affinity are often used synonymously, and this can cause confusion.

An extremely important aspect is that affinity and avidity directly influence the biological role of the antibody [2,77]. For example, the ability to facilitate antigen phagocytosis and the ability to activate the complement system contribute fundamentally to pathogen elimination and this ability is directly associated with antibody avidity [2,77]. On the other hand, the avidity of the antigen–antibody interaction is also associated with the severity of autoimmune diseases [77,78]. In addition, avidity is a parameter that directly influences immunodiagnostic reactions, including avidity measurements being used to assess the stage of a given pathology [79,80,81,82,83].

Several methodologies have been developed for the assessment of antibody affinity and avidity [1]. These methodologies can be grouped into the solution-phase, solid-phase, and equilibrium titration ELISA methodologies. Affinity/avidity determinations via solution-phase assays cover reactions where antigen and antibody interactions occur in the solution and the free antigen concentration is determined [84]. In solid-phase methodologies, the antigen is bound to a support, and after the formation of the antigen–antibody complex, the amount of antibody bound to the immune complex is determined [85]; whereas in equilibrium titration ELISA determinations, the amount of free antibody present in a solution where the immune complex formation occurs is determined [86]. The aforementioned methodologies involve the calculation of the association constant at equilibrium (K), a measure of the affinity of an antibody derived from the relationship between the concentration of the formed antigen–antibody complex and the concentrations of the antigen and free antibodies [1]. In addition to calculating the association constant, the affinity can also be defined using the dissociation constant Kdiss, as determined via the reciprocal of K (Kdiss = 1/K). Another way to evaluate the affinity/avidity of the antibody is the determination of the affinity index (AI), which is obtained using the ratio between the absorbances (Abs) arising from the antigen–antibody complex in the presence and absence of a chaotropic agent. Chaotropic agents are molecules that can disrupt the network of hydrogen bridges between water molecules and reduce the stability of the native state of the protein by reducing the hydrophobic effect [87]. The affinity index has a direct correlation with affinity [88]. 

In studies on IgY antibody avidity, methodologies that use chaotropic agents are the most commonly used [89,90,91,92]. These methodologies vary in the type of chaotropic agent used, either determining the avidity index from the ratio of the optical density obtained in the presence and absence of the chaotropic agent, or from the reduction in the optical density obtained from the use of increasing concentrations of the chaotropic agent. As in mammals, the establishment of standards for the determination of IgY antibody avidity via ELISA is extremely important [93].

## 5. Factors Affecting IgY Antibody Avidity

Like mammalian antibody avidity, chicken antibody avidity is a trait of great interest and is directly related to the development of the humoral immune response. In birds, the dynamics of the humoral immune response to an antigen are similar to those observed in mammals [94]. Initially, there is an increase in the antibody levels within 8–10 days, followed by a significant drop in antibody levels. With the administration of booster doses, an increase in the antibody levels is observed [4,95]. 

Several factors can affect the antibody production in birds and mammals. 

### 5.1. Adjuvants 

One factor is the use of substances that enhance the magnitude and durability of antibody production. These substances are called adjuvants [96]. For the production of IgY antibodies in birds, the most frequently used adjuvants are complete and incomplete Freund’s adjuvants. The primary immune response is profoundly affected by the use of Freund’s adjuvant. The use of Freund’s complete adjuvant (FCA) causes a first increase in antibody production between days 7 and 21, and a further increase in antibody production between days 42 and 59 of the initial inoculation [97]. It is interesting to note that this two-phase response stimulated via FCA also occurs in relation to the avidity of the antibodies produced, with the antibodies produced in the second phase having higher avidity than those in the first phase [97]. This effect of FCA appears to be dependent on the route of inoculation, since an intramuscular inoculation of the antigen is associated with the adjuvant results as an increase in the avidity of the antibodies produced, whereas an intravenous inoculation without the adjuvant does not lead to a significant increase [98,99]. It is important to note that in mammals, an intravenous inoculation of the antigen without adjuvant leads to a significant increase in the avidity of the antibodies produced, suggesting significant differences in the affinity maturation process between birds and mammals [98].

It is interesting to note that FCA stimulates greater avidity than other adjuvants, including FIA. A study comparing the effect of adjuvants FCA, ABM-N/-S, Gerbu, and Titer Max on IgY antibody production and avidity showed that the use of FCA results in higher avidity than the other adjuvants [76]. Another study comparing the effect of FCA and Emulsigen-D adjuvant also showed the production of antibodies with higher avidity with the use of FCA [100]. On the other hand, this effect of FCA on avidity may be related to time, since it has been observed that the use of FCA results in a faster increase in avidity compared to the use of Freund’s incomplete adjuvant or Hunter’s Titer Max adjuvant; however, at the end of the immunisation period, the avidity obtained was similar when comparing the three adjuvants [101]. In addition, the ISA VG71 adjuvant was found to have a similar effect to Freund’s adjuvants with respect to the avidity of the antibody against bothropic venom [102].

### 5.2. Time 

Regarding the time, high avidity rates (60 to 75%) are observed within 30 days after the first immunisation [101,102,103,104,105,106], and in some studies, 100% avidity rates are observed between day 7 and 21 of the first immunisation [107,108]. On the other hand, other studies did not obtain antibodies with a high avidity index (below 60%) in this same period of time [109,110,111,112,113]. In addition, some studies were not able to produce antibodies with high avidity [114,115]. It is interesting to note that, in general, avidity increases throughout the immunisation period and remains high [91,100,101,102,108,116]; however, some studies have shown a reduction in avidity after the last immunisation [104,117].

### 5.3. Other Factors 

In addition to the use of adjuvants and the timing of the immunisation, other factors, such as antigen composition, genetics, and the presence of natural antibodies, can influence the avidity of IgY antibodies. 

Studies using carrier-bound peptides show that the carrier used has an effect on the avidity of the antibody produced. Comparisons of the use of beta-lactoglobulin and KLH carriers for the production of anti-insulin antibodies showed that the inoculation of the insulin–KLH complex results in IgY antibodies with higher avidity than the application of the insulin–beta–lactoglobulin complex [118]. The use of KLH or BSA as a carrier for cancer 15-3 antigen peptides seems to influence the avidity of the IgY antibodies obtained, with the use of BSA as a carrier being related to the obtention of antibodies of higher avidity than the use of KLH, with this effect being specific to peptide 1066-1085 [116].

Genetic selection seems to be able to influence IgY antibody avidity. In an experiment selecting animals for the high and low levels of natural anti-KLH antibodies, it was observed that the serum of animals selected for the high levels of anti-KLH AcNs have IgY anti-KLH AcNs with higher levels of avidity than the AcNs of animals selected for the low levels of anti-KLH AcNs, with this effect being specific for the KLH antigen [119]. The animals selected for high SRBC antibody production have higher levels of anti-ovalbumin and anti-KLH natural antibodies (NCAs), and these antibodies have a higher avidity index than the same NCAs from the animals selected for the low anti-SRBC antibody production [120]. In both studies, the observed effect on avidity was influenced by the antigen analysed [119,120].

Furthermore, inoculation via intramuscular, intradermal, and subcutaneous routes and the dose of the antigen do not seem to influence the avidity of the antibodies obtained [112].

## 6. Comparison of Avidity of Avian and Mammalian Antibodies 

Few studies have compared antibody avidity in birds and mammals. In one study the authors obtained Kd values of 1 × 10^−12^ mol/L in birds and Kd 7 × 10^−13^ mol/L in guinea pigs [121]. In another study, K values of 1.3 × 10^10^ L/mol and 3.1 × 10^10^ L/mol were observed in birds and sheep, respectively [15]. An interesting result was found when the avidity was followed by a long immunisation process. In this study it was observed that after the first immunisation, the avidity of antibodies in birds was higher (4.7 × 10^9^ L/mol) than in sheep (5.9 × 10^8^ L/mol), but after the fourth immunisation, the avidity levels increased only 2-fold in birds and 60-fold in sheep [15]. On the other hand, other studies have observed a higher avidity of IgY antibodies towards mammals. K values ranging from 0.3 × 10^5^ M^−1^ to 15.6 × 10^6^ M^−1^ for IgY antibodies and from 0.6 × 10^5^ M^−1^ to 9.2 × 10^5^ M^−1^ for rabbit IgG antibodies have been observed [122]. Similar results were obtained in the comparison of chicken IgY and cow IgG antibodies against Escherichia coli antigen K99 [123], as well as chicken IgY and rabbit IgG anti-progesterone antibodies [124] and anti-HER and anti-human telomerase IgY antibodies in relation to rabbit IgG and mouse IgG (monoclonal) anti-HER antibodies and mouse IgG (monoclonal) anti-telomerase antibodies [125], respectively. In addition, other studies did not observe significant differences between birds and rabbits [76,101,126]. In relation to the comparison between IgY antibodies from laying hens and from rabbits, the values of Kd 2.6 × 10^−8^ and Ka of 0.478 × 10^8^ M^−1^ for IgY antibodies and Kd of 2.5 × 10^−8^ and Ka of 0.39 × 10^8^ M^−1^ for rabbit IgG antibodies have been observed [126]. Considering the possibility that the differences observed in these studies are due to the differences in species, strains, sex, and immunisation protocols, as well as the types of animals, further studies are needed to demonstrate that IgY antibodies values close to mammalian avidity can be obtained. This is especially relevant in studies on immunoprophylaxis and immunotherapy with IgY antibodies.

## 7. Methodology for the Determination of Affinity and Avidity of IgY Antibodies

Most studies on antibody affinity and avidity utilise solid phase methodologies. These studies assess IgY antibody avidity by calculating the AI using urea, magnesium chloride, or ammonium thiocyanate as the chaotropic agent. A concentration of 6 M of urea is the most commonly used. However, there is a great diversity of methodologies where the incubation time and the buffer solution of the chaotropic agent vary. Some studies use incubation for 5 or 10 min with 6 M of urea in PBS-Tween [100,106,107,116,117,127,128]; others use 6 M of urea [91,108] or 6 M of urea in PBS [111], or 6 M of urea in buffered saline [90,129]. Other studies use 6 M of urea in PBS-Tween only at the time of washing after the addition of the IgY samples [103,105]. In addition, other concentrations of urea can be used, such as 1 M [114] and 8 M [110]. For the use of magnesium chloride, two conditions are observed: incubation for 30 min after the incubations with the antibody samples [109,112,115], or the addition of magnesium chloride together with the antibody sample of magnesium chloride [101]. Ammonium thiocyanate was used in only one study, which determined the AI as the molarity of ammonium thiocyanate required for a 50% reduction in optical density relative to optical density without ammonium thiocyanate [76].

In addition to these methodologies, other solid phase methodologies have been used to assess IgY antibody avidity using the ELISA reactions [119,123], protein assay [130,131], or detection via technologies such as surface plasmon resonance [132,133] or a layered peptide array [125].

With regard to solution phase methodologies, most papers utilised radioimmunoassay reactions to assess IgY avidity [14,15,121,124,134,135]. However, indirect ELISA [120] or a fluorescence reaction have also been used [122], with the characteristic that in the vast majority of them, K or Kdiss values of IgY antibodies have been obtained. Another way to obtain an estimate of affinity is the ABC test, where the labelled antigen is incubated [97]. Finally, the least commonly utilised type of methodology to assess IgY antibody avidity is equilibrium titration ELISA [16,118,126]. In two of these studies, K or Kdiss values were obtained [16,126].

## 8. Conclusions

IgY antibodies have an affinity and avidity comparable to IgG antibodies produced by mammals. However, the processes and factors involved in the affinity/avidity maturation of IgY antibodies in birds are poorly understood. The number of studies on this topic is small. These studies show that affinity/avidity maturation is influenced by the type of adjuvant used, the number of antigen doses, the dose interval, the characteristics of the antigen, and the animal used. It is interesting to note that most studies use the determination of the avidity index via ELISA, probably due to its low cost and simplicity. However, there is great variability in the methodologies used, making it difficult to compare the results and identify the factors involved in affinity/avidity maturation accurately. Considering that these variables directly influence antibody action, it is crucial to develop a widely adopted ELISA methodology for determining avidity in IgY antibody production research. This would greatly facilitate the development of solutions in immunotherapy and immunodiagnosis based on IgY antibodies.

## Data Availability

This review did not provide new data.

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
