# Peer review of "IgY Antibodies from Birds: A Review on Affinity and Avidity"

_animals, 2023, doi:10.3390/ani13193130_

Round 1
Reviewer 1 Report
In this manuscript, the authors systematically review the studies that have investigated factors that affect the of affinity and avidity of IgY antibodies and the methodologies used to determine these variables. This work provides new insight and opinion into the affinity and avidity of IgY antibodies. The manuscript is well-organized and clearly stated. I would suggest accepting it after the following minor concerns are addressed.
First of all, the portion of the Introduction, there is an extra period in this sentence,“These antibodies. are equivalent to mammalian IgG antibodies and are the most abundant in serum and their levels increase as the humoral immune response develops”. In the second place, it is a little superfluous to describe the difference in IgY antibody between serum and yolk. Last but not the least, the portion of Conclusion is a little short and messy.
The grammar and structure of this manuscript are generally good, In addition, there are almost no spelling mistakes. I would just like to make a small suggestion.
In the portion of conclusion, it is better to delete the second "by" in the sentence "This maturation is influenced by the adjuvant used and by several other factors".
Author Response
Comments and Suggestions for Authors
In this manuscript, the authors systematically review the studies that have investigated factors that affect the of affinity and avidity of IgY antibodies and the methodologies used to determine these variables. This work provides new insight and opinion into the affinity and avidity of IgY antibodies. The manuscript is well-organized and clearly stated. I would suggest accepting it after the following minor concerns are addressed.
Answer: Thank you to the reviewer for all the comments made. They will serve to improve the manuscript. Changes made to the manuscript are indicated in blue and bold.
1) Comments: First of all, the portion of the Introduction, there is an extra period in this sentence,“These antibodies. are equivalent to mammalian IgG antibodies and are the most abundant in serum and their levels increase as the humoral immune response develops”.
Answer: The authors are grateful for the observation made by the reviewer. The text was changed by “These antibodies are equivalent...”
2) Comments: In the second place, it is a little superfluous to describe the difference in IgY antibody between serum and yolk.
Answer: The authors agree with the reviewer that the comparison between IgY in yolk and serum is superfluous. There are several articles showing that there are no significant differences. However, it is the subject of doubt for many interested people and beginners in the area. Therefore, the authors suggest maintaining the text.
3) Comments: Last but not the least, the portion of Conclusion is a little short and messy.
Answer: The authors agree with the referee's suggestions. The text has been revised to enhance its quality, and the modifications are presented below.
IgY antibodies have affinity and avidity comparable to IgG antibodies produced by mammals. However, the processes and factors involved in the affinity/avidity maturation of IgY antibodies in birds are poorly understood. The number of studies on this topic is small. These studies show that affinity/avidity maturation is influenced by the type of adjuvant used, the number of antigen doses, the dose interval, the characteristics of the antigen, and the animal used. It is interesting to note that most studies use the determination of the avidity index by ELISA, probably due to its low cost and simplicity. However, there is great variability in the methodologies used, making it difficult to compare results and identify the factors involved in affinity/avidity maturation accurately. Considering that these variables directly influence antibody action, it is crucial to develop a widely adopted ELISA methodology for determining avidity in IgY antibody production research. This would greatly facilitate the development of solutions in immunotherapy and immunodiagnosis based on IgY antibodies.
Comments on the Quality of English Language
The grammar and structure of this manuscript are generally good, In addition, there are almost no spelling mistakes. I would just like to make a small suggestion.
Answer: Thank you for your comments.
1) Coments: In the portion of conclusion, it is better to delete the second "by" in the sentence "This maturation is influenced by the adjuvant used and by several other factors".
Answer: Dear reviewer, we have removed the word "by".
Reviewer 2 Report
Authors tried to prepare a review manuscript that provides an insight on the significant subject of IgY antibodies from birds: a review on affinity and avidity. However, some issues need further clarification before acceptance.
Authors must explain the physiological significance of this evaluation- this not clear in the introduction nor in the aim of this literature review. Second, authors did not expalin the relationship of affinity, and avidity with the antibody titers. In my opinion, this is a relationship that audience may understand in case there is a correlation. Authors, did not comment on lipid oxidation, stress conditions- heat stres or other stresses that may interefere such as the presence of heat schock proteins, apoptotic proteins, CD and occludin proteins. Also, inline 157-158 dark and light zone is not clear to reader. Finally, conclusion need rephrasing, in order to provide some clear suggestions and not confusing remarks.
The use of english are acceptable. I read the manuscript carefully, however, I did not spotted any mistake or mistyping.
Author Response
Comments and Suggestions for Authors
Authors tried to prepare a review manuscript that provides an insight on the significant subject of IgY antibodies from birds: a review on affinity and avidity. However, some issues need further clarification before acceptance.
Answer: Thank you to the reviewer for all the comments made. They will serve to improve the manuscript. Changes made to the manuscript are indicated in blue and bold.
1) Comments: Authors must explain the physiological significance of this evaluation- this not clear in the introduction nor in the aim of this literature review.
Answer: Thank you for your observation. It will help improve the text. In relation to your observations, it is important to note that avidity and affinity are essential variables for the function of antibodies. They directly influence the binding time of the antibody with the antigen and, consequently, the biological functions of the antibodies. Despite the importance of these variables, they are little investigated in birds. This has a great impact on the development of solutions for immunotherapy and immunodiagnosis based on IgY antibodies. Understanding the factors that directly influence the production of IgY antibodies with greater avidity will certainly contribute to the establishment of immunization protocols that result in the production of IgY antibodies with greater avidity in a shorter time. This will represent a reduction in the cost of producing these antibodies, IgY, and consequently increasing the competitiveness of these antibodies in relation to those produced in mammals. Therefore, this review will contribute significantly to reflection on this topic. In an attempt to make the objective of this review explicit, the introduction was changed as described below:
Currently, IgY antibodies have been used to develop applications for immunotherapy and immunodiagnosis of diseases in humans and animals [5]. Despite their many advantages over mammalian antibodies, there are few IgY-based products available on the market. Understanding the mechanisms involved in affinity maturation can result in the establishment of immunization protocols that lead to the production of high-avidity IgY antibodies in the shortest possible time. Consequently, this can increase the competitiveness of these antibodies compared to those produced in mammals.
2) Comments: Second, authors did not expalin the relationship of affinity, and avidity with the antibody titers. In my opinion, this is a relationship that audience may understand in case there is a correlation.
Answer: The authors thank you for your comment. As far as we know, the antibody titer and avidity/affinity variables are not directly related. What is generally observed is that there is an increase in antibody titers until stabilization with repeated immunizations, while with avidity/affinity, it can stabilize or continue to increase after repeated immunizations. An extremely important aspect that needs to be better studied is what factors lead to the stabilization or continuous increase in observed avidity. Are these factors that can be manipulated in favor of the increase?
3) Comments: Authors, did not comment on lipid oxidation, stress conditions- heat stres or other stresses that may interefere such as the presence of heat schock proteins, apoptotic proteins, CD and occludin proteins.
Answer: Thanks for the observation. The reviewer is completely right in pointing out these variables as important in the maturation of the avidity/affinity of IgY antibodies. Unfortunately, there are no studies on the impact of these variables on the avidity/affinity of IgY antibodies. Therefore, no comments were made on these variables in the text.
4) Comments: Also, inline 157-158 dark and light zone is not clear to reader.
Answer: The authors agree with the reviewer. The text has been changed as described below.
The dark zone is an area within the germinal center where numerous B lymphocytes are actively dividing and undergoing somatic hypermutation. In contrast, the light zone contains fewer cells and is responsible for stimulating the survival of B lymphocytes through various processes.
5) Comments: Finally, conclusion need rephrasing, in order to provide some clear suggestions and not confusing remarks.
Answer: The authors agree with the referee, and changes were made to improve the text. The following paragraphs describe the changes made to the completion.
IgY antibodies have affinity and avidity comparable to IgG antibodies produced by mammals. However, the processes and factors involved in the affinity/avidity maturation of IgY antibodies in birds are poorly understood. The number of studies on this topic is small. These studies show that affinity/avidity maturation is influenced by the type of adjuvant used, the number of antigen doses, the dose interval, the characteristics of the antigen, and the animal used. It is interesting to note that most studies use the determination of the avidity index by ELISA, probably due to its low cost and simplicity. However, there is great variability in the methodologies used, making it difficult to compare results and identify the factors involved in affinity/avidity maturation accurately. Considering that these variables directly influence antibody action, it is crucial to develop a widely adopted ELISA methodology for determining avidity in IgY antibody production research. This would greatly facilitate the development of solutions in immunotherapy and immunodiagnosis based on IgY antibodies.
Comments on the Quality of English Language
The use of english are acceptable. I read the manuscript carefully, however, I did not spotted any mistake or mistyping.
Answer: Thank you for your comments.
Reviewer 3 Report
The review entitle “ IgY antibodies from birds: a review on affinity and avidity” is well written and well cited, however I have some comments/suggestions.
Line 27: what do you mean by strain of the animal? is that the species?
Line 141: the sentence " These forces are stronger the smaller the distance between the elements .." please rewrite/revise. It should be “the stronger the forces, the smaller the distance….”
Line191: Several methodologies have been developed for the assessment of antibody affinity and avidity..... : I suggest to move this to “6. Methodology for the determination of affinity and avidity of IgY antibodies (Line 312).
Line 219: the title “5. Factors affecting IgY antibody avidity” , I suggest to add subtitles of these factors.
References do not follow the journal guidelines, please revise.
English is fine
Author Response
Comments and Suggestions for Authors
The review entitle “ IgY antibodies from birds: a review on affinity and avidity” is well written and well cited, however I have some comments/suggestions.
Answer: Thank you for all the comments made. They will serve to improve the manuscript. Changes made to the manuscript are indicated in blue and bold.
1) Comments: Line 27: what do you mean by strain of the animal? is that the species?
Answer: Thank you for your observation. The authors should use the word "lineage" instead of the word "strain." There are different lineages of broilers and laying hens, such as Coob and Roos308 for broilers, and Hy-line and Isa Brown for laying hens. The text has been modified accordingly.
“..use of an adjuvant, type of antigen, animal lineage, number of immunisations,..”
2) Comments: Line 141: the sentence " These forces are stronger the smaller the distance between the elements .." please rewrite/revise. It should be “the stronger the forces, the smaller the distance….”
Answer: Thank you for your observation. The text has been revised to improve clarity.
“...van der Walls forces [62, 63]. These forces are stronger when the distance between the elements is smaller. Therefore,..”
3) Comments: Line191: Several methodologies have been developed for the assessment of antibody affinity and avidity..... : I suggest to move this to “6. Methodology for the determination of affinity and avidity of IgY antibodies (Line 312).
Answer: Thank you for your observation. However, the authors firmly believe that this sentence serves as a crucial introduction to the subsequent text. Therefore, we strongly believe it is more appropriate to retain the text in its current form.
4) Comments: Line 219: the title “5. Factors affecting IgY antibody avidity” , I suggest to add subtitles of these factors.
Answer: The authors agree with the reviewer' feedback. The subtitles have been added as described below.
“5.1. Adjuvants...”
“5.2. Time...”
“5.3. Others factors...”
5) Comments: References do not follow the journal guidelines, please revise.
Answer: Thank you for the observation made. The references were reviewed, and necessary changes were made. Considering the large number of changes made, they have not been described here.
Comments on the Quality of English Language
English is fine
Answer: Thank you for your comments.
Reviewer 4 Report
I have some minor comments:
1. Provide some more references of immunotherapy use of IgY for bacterial, viral, fungal and parasites in line# 98-99.
2. Provide some 1-2 line explanation/definition of Kd and K value and its importance so that its easier for reader to understand the values, line 288-289.
3. Line 338 ‘titration ELISA ) remove the bracket.
4. Line 428 ref 31 , e is missing in egg yolk antibodies.
Author Response
Answer: Thank you for all the comments made. They will serve to improve the manuscript. Changes made to the manuscript are indicated in blue and bold.
1) Comments: Provide some more references of immunotherapy use of IgY for bacterial, viral, fungal and parasites in line# 98-99.
Answer: The authors are grateful for the suggestion made by the reviewer. Some references have been included.
“...In particular, IgY antibodies have been studied for immunotherapy of bacterial [12 ,33], viral [29, 34], fungal [30, 35], parasitic [31, 36], respiratory [37], enteric [38, 39, 40], and chronic diseases, such as periodontitis, cystic fibrosis, and coeliac disease [41, 42, 43].”
2) Comments: Provide some 1-2 line explanation/definition of Kd and K value and its importance
so that its easier for reader to understand the values, line 288-289.
Answer: The authors agree that these definitions are extremely important. Therefore, a description of their meanings can be found in lines 155-156, 187-188, and 207-213, and they are in bold in the manuscript.
“...Affinity can be expressed by the association constant at equilibrium (K) or by the dissociation constant (Kd or Kdiss) [1]….”
“... Avidity can also be expressed in terms of the constants K and Kd [1]….”
“...The aforementioned methodologies involve the calculation of the association constant at equilibrium (K), a measure of the affinity of an antibody derived from the relationship between the concentration of the formed antigen-antibody complex and the concentrations of antigen and free antibodies [1]. In addition to calculating the association constant, the affinity can also be defined by the dissociation constant Kdiss, determined by the reciprocal of K (Kdiss= 1/K)...”
3) Comments: Line 338 ‘titration ELISA ) remove the bracket.
Answer: The authors thank you for your observation. The text has been changed as suggested by the reviewer.
“...equilibrium titration ELISA [16, 118, 126]…”
4) Comments: Line 428 ref 31 , e is missing in egg yolk antibodies.
Answer: The authors thank you for your observation. The text has been changed as suggested by the reviewer.
“...Egg yolk antibodies (IgY)...”
Round 2
Reviewer 2 Report
Authors revised adequately.